# Role of Remittance on Sustainable Economic Development in Developing and Emerging Economies: New Insights from Panel Cross-Sectional Augmented Autoregressive Distributed Lag Approach

**Shasnil Avinesh Chand** [1,2,*]  **and Baljeet Singh** [1]

[1] School of Accounting, Finance and Economics, The University of the South Pacific, Laucala Campus, Laucala Bay Road, Suva, Fiji; singh_bl@usp.ac.fj

[2] School of Economics and Finance, Fiji National University, Suva P.O. Box 3722, Fiji

[*] Correspondence: shasnil.chand@fnu.ac.fj

**Abstract:** In this study, we aim to investigate the effects of remittance on sustainable economic development in 52 developing and emerging economies from 1996 to 2021. The study uses other variables such as real GDP per capita, total natural resource rents, globalization, and foreign direct investment. To achieve the mentioned objective, we apply a series of second-generation panel estimation approaches. These include CIPS unit root, Westerlund cointegration, cross-sectional augmented autoregressive distributed lag (CS-ARDL), and robustness using augmented mean group (AMG) and common correlated mean group (CCEMG). These methods are useful provided they are robust towards cross-country dependencies, slope heterogeneity, endogeneity, and serial correlation, which are disregarded in the conventional panel estimations. The empirical findings indicate that remittance accelerates sustainable economic development. Additionally, real GDP per capita and globalization also positively contribute towards sustainable economic development. However, total resource rents deteriorate sustainable economic development. This study offers key policy implications based on the empirical findings for the developing and emerging economies.

**Keywords:** remittance; sustainable economic development; CS-ARDL; AMG; CCEMG

## 1. Introduction

Sustainable economic development (SED) entails the utilization and management of resources in a manner that preserves the needs of future generations, thereby preventing the depletion of national capital (Güney 2019). Countries across the globe have increasingly adopted SED as a guiding framework for promoting sustainable growth that is beneficial to human development and environment quality in the long term (Hunjra et al. 2022a). This involves harmonizing and integrating natural, social, human, and financial capital to achieve a balanced and sustainable approach. Advocates of SED argue that developing countries can attain sustainability by simultaneously minimizing the exploitation of natural resources and increasing investment in human and physical capital. Therefore, this study examines the role of remittances on sustainable development in developing and emerging economies.

Remittances, constituting important component of the financial landscape with nearly USD 600 billion per year flowing into low- and middle-income countries, hold considerable potential for bolstering sustainable economic development within developing and emerging economies. The inflow of remittance funds serves as a vital source of income for families, aiding in poverty alleviation (Acheampong et al. 2021; Azam et al. 2016; Xia et al. 2022). The impact of remittances extends beyond individual households to influence the overall economic landscape of recipient economies. These financial inflows contribute significantly

to the gross domestic product (GDP) of many developing and emerging economies (Barajas et al. 2009; Meyer and Shera 2017).

The infusion of remittance funds into national economies has the potential to stimulate economic growth and development. Governments in recipient countries can leverage these inflows to invest in infrastructure, healthcare, education, and green projects, fostering an environment conducive to sustainable development. Additionally, remittances act as a stabilizing force during times of economic downturns, providing a counterbalance to external shocks and uncertainties. Moreover, these funds can be directed towards sustainable and climate-resilient projects, contributing to effective climate change management.

On the flip side, remittances can contribute to environmental threats, specifically the surge in carbon emissions. The boost in people's purchasing power facilitated by remittances enables the acquisition of luxury items such as vehicles and other transportation goods. Subsequently, these vehicles and machinery consume energy through fuel combustion, leading to a significant release of carbon dioxide into the atmosphere (Qiao et al. 2024; Sharma et al. 2019; Wang et al. 2021; Chen et al. 2022). This rapid escalation in carbon dioxide emissions accounts for 58.5% of greenhouse gas emissions, playing a pivotal role in climate change and global warming (Halicioglu 2009).

Despite the extensive body of research examining the relationship between remittances and economic growth, encompassing both country-specific and cross-country analyses, there remains a limited exploration of the relationship between remittances and SED. The existing literature has predominantly focused on the remittance and growth nexus, yet a comprehensive understanding of how remittances contribute to long-term, sustainable economic development is still lacking. Subsequently, it is important to analyze the contribution of remittance on SED.

Accordingly, this study aims to examine the impact of remittance on SED for a large panel of 52 countries over the period of 1996–2021. In this study, we have used adjusted net savings (ANS) from the World Bank as a proxy for sustainable economic development. ANS is a recently developed proxy for assessing economic sustainability (Güney 2019; Hunjra et al. 2022a; Hunjra et al. 2023; Hussain et al. 2023). It assesses whether a country's savings and investments adequately offset the depreciation and depletion of physical and natural capital as well as damages caused by pollution. Adjusted net savings are calculated by making four adjustments to the national accounting measure of gross saving: (a) deducting the consumption of fixed capital to derive net national savings; (b) adding government education expenditure on education to consider investment in human capital; (c) subtracting estimates of the depletion of various natural resources to account for the decrease in asset values linked to extraction and depletion; and (d) making deductions for damages resulting from carbon dioxide and particulate emissions associated with extraction and depletion. Theoretically, remittances can contribute to ANS (SED) through various channels. Firstly, remittance inflows can promote economic growth by stimulating domestic consumption and investment (Glytsos 2005), thus resulting in higher gross savings and investment. Secondly, remittance contributes to higher private and public sector investment in education and other forms of human capital accumulation (Azizi 2018). Thirdly, remittance can contribute to the reduction in carbon emissions and more sustainable development projects (Ahmad et al. 2022). In the context of the above theory and discussion, we hypothesize that remittances may contribute to SED in remittance-receiving countries.

This study contributes to the extant literature at least in two major ways; first, this study is one of the first study to examine the impact of remittances on adjusted net savings, which are widely used as a measure of sustainable economic development (see Güney 2019; Hunjra et al. 2022b, 2023; Hussain et al. 2023). An analysis of the impact of remittances on ANS will offer valuable insights into how remittances contribute to the productive assets essential for meeting the economic needs of future generations. Second, digressing from conventional estimation methods employed in previous studies, this research adopts a second-generation econometric technique, specifically CS-ARDL, an innovative dynamic panel approach. This choice is made to address issues of cross-sectional dependence and

the heterogeneity of slope coefficients. The utilization of CS-ARDL, particularly in the presence of weak endogeneity, enhances the robustness of results compared to conventional panel data estimators. Our findings exhibit that remittance helps to increase sustainable economic development. Additionally, real GDP per capita and globalization also positively contribute towards sustainable economic development. However, total resource rents deteriorate sustainable economic development.

The rest of the paper is structured as follows: Section 2 reviews the relevant literature. Section 3 describes the data and methodology. Section 4 presents the results and discussion. Section 5 concludes the paper with some key policy recommendations.

## 2. Literature Review

*Linking Remittance and SED*

The nexus between remittances and economic growth is widely studied in academic literature; however, empirical evidence is mixed. In a meta-analysis, Cazachevici et al. (2020) reviewed 538 estimates from 95 studies and noted that around 40% of the studies observed a positive effect, 40% found no significant effect, and 20% of the studies found evidence of negative impact. The findings suggest a publication bias in favor of positive effects. Upon correcting for this bias using advanced techniques, the study concludes that while the mean effect of remittances on growth remains positive, it is economically modest. In another bibliometric and systematic literature review, Radic et al. (2023) found that a large quantity of published papers did not come from countries with high share of remittances to GDP. These findings suggest a positive correlation between remittances and economic growth, with the nature of this relationship differing based on country's income level.

Despite the empirical inconclusiveness, remittance inflows can stimulate the economic growth and development of the developing economies through various channels. For instance, remittances can help to improve poverty alleviation by directly benefiting recipient households (Azam et al. 2016; Masron and Subramaniam 2018; Peković 2017). Families in developing countries often rely on remittances for basic necessities such as food, housing, healthcare, and education (Kumar et al. 2018). This influx of funds enhances the welfare and living standards of recipients, thereby reducing poverty levels and promoting economic stability. The findings of Masron and Subramaniam (2018) generally supported the notion that higher inflows of remittances were associated with a decrease in poverty levels in 44 developing countries. This is attributed to the fact that increased remittance flows are often directed towards more productive activities, thereby contributing to sustainable poverty reduction efforts. Subramaniam et al. (2023) found that remittances help to alleviate energy poverty, facilitate energy access, and enhance energy security in 50 developing countries. Hosan et al. (2023), using household income and expenditure survey in Bangladesh, found that an increase in remittance has a positive impact on alleviating energy poverty. González Bautista et al. (2024) found that economic growth and financial development acted as mediators, allowing remittances to indirectly contribute to reducing energy poverty in Latin American countries.

Remittance inflows can stimulate domestic consumption and investment (Glytsos 2005). The additional income received through remittances fuels consumption spending, driving demand for goods and services in local markets (Zarate-Hoyos 2004). This increased consumer spending, in turn, spurs business activity and stimulates economic growth. Moreover, remittances can be used for productive investments such as starting small businesses, acquiring assets, or funding entrepreneurial ventures (Amuedo-Dorantes and Pozo 2006). These investments contribute to job creation, innovation, and overall economic productivity. Kakhkharov (2019) found that remittances stimulate entrepreneurial activities across 63 developing nations. Similarly, Alhassan (2023) revealed that employing e-government for migrant service delivery yields a favorable net association between remittances and the establishment of new formal businesses in 55 recipient economies.

Remittance inflows serve as a stable source of foreign exchange earnings for recipient economies (Ratha and Mohapatra 2007). In many developing economies, remittances

constitute a significant portion of foreign currency reserves (Guha 2013). This influx of foreign exchange helps stabilize exchange rates, improve liquidity in financial markets, and mitigate balance of payment deficits. Additionally, remittances can reduce the reliance on volatile sources of foreign exchange such as exports or foreign direct investment, thus enhancing economic resilience (Singer 2010).

Remittances can facilitate human capital development and skill transfer (Azizi 2018; Ngoma and Ismail 2013). Migrant workers often acquire valuable skills, knowledge, and experiences in host countries, which they can transfer back to their home countries upon return (Cassarino 2004). Additionally, remittance recipients may use a portion of the funds to invest in education and skills training for themselves or their children (Rapoport and Docquier 2006). This investment in human capital enhances the workforce's productivity and employability, ultimately fostering long-term economic growth and development (Blundell et al. 1999). Azizi (2018) found that remittances increased school enrollment, school completion rate, and private school enrollment in 122 developing economies. Huay et al. (2019) found that the effect of remittances was statistically significant with positive coefficients in 67 developing economies. Sahoo et al. (2020) found the positive effect of remittance on human development in South Asian countries. Ali Bare et al. (2022) examined the effect of remittances on human capital development in sub-Saharan Africa. Their study highlighted that remittances exert a positive influence on human capital investment. Xia et al. (2022) found a positive effect of remittance on human capital development for top 10 remittance-receiving economies.

Besides economic impacts, remittance inflows can also have environmental impacts. This increase in income from remittances often results in higher levels of consumption and production, including the purchase of goods and services that have carbon footprints associated with their production and transportation (Wang et al. 2021). For example, the recipients of remittances may buy more energy-intensive products such as electronics, automobiles, or luxury goods, which can contribute to higher carbon emissions. However, remittance inflows can also have the opposite effect and contribute to the reduction in carbon emissions and more sustainable development projects (Ahmad et al. 2022). This could include the installation of solar panels, energy-efficient appliances, or improved insulation in buildings, leading to a reduction in carbon emissions associated with energy consumption. Remittance inflows can support sustainable development initiatives in recipient countries (Zafar et al. 2022). This might include investments in conservation projects, reforestation efforts, or sustainable agriculture practices, which can help mitigate carbon emissions. Raihan and Voumik (2022) found that remittances help to lower carbon emissions in China. The findings of Ahmad et al. (2022) indicated that the positive shock of remittances contributes to pollution emissions, while the negative shock of remittances mitigates the pollution both in the long and short run. Karmaker et al. (2023) examined the impact of remittance on renewable energy consumption for 25 top remittance-receiving economies and found that remittance increased renewable energy consumption. Similarly, the empirical findings of Subramaniam et al. (2023) showed that remittance increased renewable energy consumption in India, China, Mexico, and Philippines.

While there are several studies that analyzed the impact of remittances on various macroeconomic variables; however, at least to our knowledge, no study examined the impact of remittance on SED using a broad indicator of SED such as ANS. This study is a modest attempt to fill this gap in the extant literature. Remittances can directly contribute to SED via its impact on human capital investment. Remittances can boost private as well as public investment in education. Similarly, remittances can contribute to physical capital investment, which is also critical for SED. Remittances can also contribute to SED through its impact on $CO_2$ emissions. Remittance is likely to facilitate the greater adaptation of green technology, hence lowering $CO_2$ emissions. Also, remittance, which is closely aligned with the movement of people across the border, is likely to reduce the exploitation of natural resources in remittance-receiving countries, which further likely boosts SED.

## 3. Data and Method

### 3.1. Data

The study uses a panel dataset of 52 developing and emerging economies over the 1996–2021 period. Following Hassan and Holmes (2013), we include high remittance-receiving countries, with an average remittance-to-GDP ratio of 1% or over at the end of our period of 1996–2021. We include annual data for the examination. For maximizing the total number of observations in a balanced panel, the sample includes Albania, Armenia, Azerbaijan, Bangladesh, Barbados, Belarus, Belize, Benin, Bolivia, Cabo Verde, Cambodia, Colombia, Costa Rica, the Dominican Republic, Ecuador, Egypt, El Salvador, Fiji, Georgia, Ghana, Guatemala, Guinea-Bissau, Haiti, Honduras, Hungary, India, Jamaica, Jordan, Kenya, the Kyrgyz Republic, Mali, Mexico, Moldova, Mongolia, Morocco, Nepal, Nicaragua, Nigeria, Niger, Pakistan, Paraguay, Peru, the Philippines, Poland, Senegal, the Solomon Islands, Sri Lanka, Togo, Tunisia, Türkiye, Vanuatu, and Uganda. The total amount of remittance received by the selected countries is around 56% of total remittance received in year 2021.[1] In Appendix A, Figure A1 displays adjusted net savings (ANS) as a percentage of gross national income (%GNI), while Figure A2 presents remittances as a percentage of gross domestic product (GDP) for individual cross-sections. For the purpose of our analysis, all used variables are transformed into their natural logarithm forms. The issue of non-linearity, heteroscedasticity, outlier, and skewness in the data can be dealt by transforming into log (Nica et al. 2023). There were instances where we had negative values for adjusted net savings and foreign direct investment. Following Kumar and Stauvermann (2023), we scaled the series using the formula: $X_t^{scaled} = \ln\left(X_t^{actual} + ABS\left(\min\left(X^{all}\right) + 0.0001\right)\right)$, where $X_t$ is either adjusted net savings or foreign direct investment. ABS denotes the absolute value, min shows the minimum (largest negative) value of either ANS or FDI, $X^{all}$ represents the entire series of either ANS or FDI, and 0.0001 is used as the correction factor.

**Table 1.** Variables description.

| Name | Symbol | Description of Variable | Data Source |
|---|---|---|---|
| Sustainable economic development | LANS | Adjusted net savings, excluding particular emission damage (% GNI). | WDI |
| Remittance | LREM | Remittance-to-GDP ratio | WDI |
| Economic growth | LRGDPP | GDP per capita (constant 2015 USD) | WDI |
| Natural resource rents | LRES | Total natural resource rents (% GDP) | WDI |
| Globalization | LGLOB | Economic, social, and political dimensions of globalization index | KOF |
| Foreign direct investment | LFDI | Foreign direct investment net inflows (% GDP) | WDI |

**Table 2.** Descriptive statistics.

| Descriptive Analysis | | | | | | |
|---|---|---|---|---|---|---|
| | LANS | LREM | LRGDPP | LRES | LGLOB | LFDI |
| Mean | 3.53 | 1.40 | 7.82 | 0.76 | 3.98 | 3.77 |
| Median | 3.56 | 1.48 | 7.87 | 0.72 | 4.01 | 3.75 |
| Maximum | 4.23 | 3.54 | 9.84 | 3.90 | 4.44 | 4.99 |
| Minimum | −4.35 | −2.43 | 5.96 | −4.68 | 3.24 | −9.21 |

**Table 2.** *Cont.*

| Descriptive Analysis | | | | | | |
|---|---|---|---|---|---|---|
| Std. Dev. | 0.42 | 1.10 | 0.88 | 1.41 | 0.22 | 0.38 |
| Skewness | −6.75 | −0.60 | 0.01 | −0.26 | −0.55 | −30.67 |
| Kurtosis | 103.47 | 3.38 | 2.35 | 2.84 | 3.09 | 1055.06 |
| Probability | 0.00 | 0.00 | 0.00 | 0.00 | 0.00 | 0.00 |
| Observations | 1352 | 1352 | 1352 | 1352 | 1352 | 1352 |

*Source*: authors' computation.

**Table 3.** Correlation matrix.

| Correlation Matrix | | | | | | |
|---|---|---|---|---|---|---|
| | LANS | LREM | LRGDPP | LRES | LGLOB | LFDI |
| LANS | 1 | | | | | |
| LREM | 0.170 | 1 | | | | |
| LRGDPP | 0.131 | −0.134 | 1 | | | |
| LRES | −0.187 | −0.208 | −0.324 | 1 | | |
| LGLOB | 0.154 | 0.030 | 0.212 | 0.063 | 1 | |
| LFDI | −0.023 | 0.018 | −0.005 | 0.037 | −0.006 | 1 |

*Source*: authors' computation.

### 3.2. Method

Following the standard literature on remittance and growth, we start from the assumption that a transfer of remittances could positively contribute to SED. If remittances are used for productive investments or contribute to the preservation of natural resources (thereby reducing environmental degradation), they could positively impact SED. Therefore, we have the following:

$$LANS = f(LREM, X) \tag{1}$$

where *LANS* represents the natural log of adjusted net savings (proxy for sustainable economic development), while *LREM* denotes worker remittances as a percent of GDP, and *X* reflects the vector of other determinants of the adjusted net savings. Regarding all other determinants of sustainable economic development, the study proposes the following:

(i) Economic growth (+/−): Conventional economic growth models often rely on the exploitational of natural resources, leading to environmental degradation and depletion (Gylfason and Zoega 2006; Zallé 2019). Conversely, economic growth can also drive innovation and technological advancements that promote sustainable practices, resource efficiency, and the development of cleaner technologies (Galindo and Méndez 2014). The existing literature predominantly focuses on the ramifications of economic growth on the environment, particularly through the lens of the Environmental Kuznets Curve hypothesis. This hypothesis posits that the initial phases of economic development tend to coincide with heightened environmental degradation (Cole and Maxwell 2003; Grossman and Krueger 1991). This degradation is often attributed to the escalated exploitation of natural resources, intensified production activities, and rapid industrialization. Consequently, environmental degradation appears to be an unavoidable consequence in regions undergoing development, where nations have either embarked on or are in the process of embarking on their developmental journey. On the other hand, the intuition is that economic activities are helpful for sustainable development through the conservation of natural resources (Hunjra et al. 2022a), investment in human and physical capital, and green technologies. Hence, the impact of economic growth on SED is ambiguous.

(ii) Natural resource rents (+/−): The exploitation of natural resources often involves environmental degradation, including deforestation, pollution, and habitat destruction, further exacerbating the sustainability challenge (Singh and Singh 2017). Moreover, while natural resource exports can initially boost economic growth, the Dutch Disease

effect can undermine the diversification and resilience needed for sustainable development, both economically and environmentally (Singh and Singh 2017). Dutch disease hypothesis (Corden 1984) argues that an increase in the export of natural resources leads to the appreciation of domestic currency, which negatively affects the productive sectors of the economy. Empirical studies such as Haseeb et al. (2021) and Khan et al. (2021) found that the positive effects of natural resource rents and sustainable development. On the contrary, Arslan et al. (2022) and Qian et al. (2021) found the negative relationship between natural resource rents and sustainable development. Hence, the impact of natural resource rent on SED is ambiguous.

(iii) Globalization (+): Globalization is defined as an increase in the integration between the markets for goods, services, and capital (Amavilah et al. 2017). Globalization facilitates the specialization of countries in industries where they have a comparative advantage (Requier-Desjardins et al. 2003). This specialization leads to increased efficiency and productivity, which can drive economic growth and development sustainably. For example, a country with abundant natural resources might specialize in resource extraction and export, while another country with a skilled workforce might specialize in high-tech manufacturing or services. Through trade, both countries can benefit from exchanging goods and services, leading to overall economic development. Globalization can incentivize countries to improve their institutional frameworks to attract investment and participate more effectively in the global economy. Studies such as Arif et al. (2022), Sethi et al. (2020), and Umar et al. (2020) found the positive effects of globalization on sustainable development. Therefore, we hypothesize that globalization has a positive effect on SED.

(iv) Foreign direct investment (FDI) (+/−): The impact of FDI on SED could be associated with pollution halo hypothesis and pollution haven hypothesis. Based on pollution halo hypothesis, FDI will positively impact environmental quality (Ahmad et al. 2021; Balsalobre-Lorente et al. 2019; Mert and Caglar 2020). The reasoning behind this is that multinational corporations (MNCs) from developed countries, which often bring foreign direct investment, may transfer their advanced technologies and management practices to the host country. These technologies and practices could include cleaner production methods, energy efficiency measures, and waste management systems. As a result, the environmental performance of the host country may improve, contributing to sustainable economic development. On the contrary, based on pollution haven hypothesis, FDI will negatively impact environmental quality (Ahmad et al. 2021; Balsalobre-Lorente et al. 2019; Mert and Caglar 2020). According to this hypothesis, MNCs may seek to invest in countries with negligent environmental regulations and enforcement to minimize costs. By doing so, they can avoid stringent environmental standards and regulations that exist in their home countries, thereby exploiting the host country's resources and contributing to pollution and environmental degradation. This could lead to negative environmental impacts such as air and water pollution, deforestation, and habitat destruction, undermining sustainable development efforts. Studies such as Dornean et al. (2021) found the positive impact of FDI on SED. Zamani and Tayebi (2022) found the negative impact of FDI on SED. However, Wang et al. (2023) and Ayamba et al. (2020) noted the insignificant impact of FDI on SED. Hence, the impact of FDI on SED is ambiguous.

$$LANS_{it} = \beta_0 + \beta_1 LREM_{it} + \beta_2 LRGDPP_{it} + \beta_3 LRES_{it} + \beta_4 LGLOB_{it} + \beta_4 LFDI_{it} + \varepsilon_{it} \qquad (2)$$

In the equation, $i$, $t$, and $\varepsilon_{it}$, respectively, signify cross-section, time period, and residual term. Furthermore, *LANS* represents adjusted net savings as a measure of sustainable economic development, *LREM* denotes remittance, *LRGDPP* refers to economic growth, and *LRES* represents natural resource rents, while *LGLOB* and *LFDI* represent globalization and foreign direct investment, respectively.

This study adopts six steps for our estimation procedure to overcome the commonly arising issues with economic techniques investigating the dynamic impact of remittance

inflow on sustainable economic development. First, the Pesaran cross-sectional dependence test is used to test dependency among the economies. Second, the slope homogeneity test is performed based on the Pesaran and Yamagata (2008). Third, the CIPS unit root test is performed to confirm the stationarity. Fourth, the panel bootstrap cointegration test is performed based on Westerlund and Edgerton (2007) to verify the long-run relationship. Fifth, the cross-section augmented autoregressive distributed lag (CS-ARDL) test is performed for determining long-run and short-run relationships. Finally, the robustness of the long-run estimation is sanctioned through AMG and CCEMG estimators.

### 3.2.1. Cross-Sectional Dependence Test

The cross-sectional dependency test is applied prior to unit root test to examine the heterogeneity in slope parameters. Conventional panel data estimation assumes that no dependency exists between cross-sectional units and slope coefficients are homogenous. However, ignoring cross-sectional dependence may cause false inferences (Chudik and Pesaran 2013). Given the impact of globalization, the interconnection amongst the developing and emerging economies is possible. Subsequently, estimating cross-sectional dependence among these nations is important, and ignoring them will lead to inconsistent and ambiguous outcomes. For the purpose of this study, we employ (Pesaran 2004) the cross-sectional dependence test as this test is valid for large $N$ and $T$, and the equation can be expressed as follows:

$$CD = \sqrt{\frac{2K}{N(N-1)}} \sum_{i=1}^{K-1} \sum_{j=i+1}^{K} T_{ij} \hat{\varphi}_{ij}^2 \rightarrow N(0,1), K = 1, 2, 3, \ldots . N \quad (3)$$

where $K$, $T$, and $\hat{\varphi}_{ij}^2$ denote sample size, time period, and correlation of residuals between country $i$ and $j$, respectively.

### 3.2.2. Slope Homogeneity Test

Misleading estimates may arise when there is slope homogeneity in a panel that is inherently heterogeneous. Hence, it is essential to address cross-sectional heterogeneity during empirical investigations. Subsequently, we employ the slope homogeneity test of Pesaran and Yamagata (2008) since it fits well with large $N$ and $T$ in the panel. The equations for Pesaran and Yamagata (2008) test statistics are as follows:

$$SP = \sum_{i=1}^{N} (\sigma_i - \sigma_W) \frac{x_i M_\varphi x_i}{\vartheta_i^2} (\sigma_i - \sigma_W) \quad (4)$$

$$\Delta = \sqrt{N \left( \frac{N^{-1} S - L}{\sqrt{2L}} \right)} \quad (5)$$

$$\Delta_{adj} = \sqrt{N} \left( \frac{N^{-1} S - L}{\sqrt{\frac{2L(T-L-1)}{(T+1)}}} \right) \quad (6)$$

In the equation, $\sigma_i$ = coefficient of pooled OLS; $\sigma_W$ = weighted pooled fixed effect; $x_i$ = matrix of independent variables derived from mean deviations; $M_\varphi$ = determine matrix; $\vartheta_i^2 = \vartheta_i$ estimate; $SP$ and $\Delta$ are test statistics; and $L$ = number of regressors. $\Delta_{adj}$ = adjusted form of $\Delta$.

### 3.2.3. Panel Unit Root Test

In the case of cross-sectional dependence, the conventional panel unit root test such as Phillpps–Perron (Fisher-PP); augmented Dickey–Fuller (Fisher-ADF); Im, Peasran, and Shin (IPS); and Levin–Lin–Chu (LLC) become inappropriate. Subsequently, this study applies the second-generation panel unit root test proposed by Pesaran (2007). The Pesaran CIPS unit root test ensures the consideration of averages across cross-sections, lagged values,

and the first difference for cross-section augmentation. The equation for the CIPS unit root test is as follows:

$$CIPS = \frac{1}{N}\sum_{i=1}^{N} t_j(N, T)$$

(7)

### 3.2.4. Panel Cointegration Test

The growing empirical literature emphasizes the importance of panel cointegration test to confirm the long-run relationship amongst the variables. For the purpose of this study, we employ the Westerlund cointegration test proposed by Westerlund and Edgerton (2007). This test is appropriate in the presence of cross-sectional dependence and slope heterogeneity. The equation for Westerlund cointegration test contains the following:

$$G_t = \frac{1}{N}\sum_{j=1}^{N} \frac{\theta_j^t}{SE\theta_j^t}$$

(8)

$$G_\alpha = \frac{1}{N}\sum_{j=1}^{N} \frac{T_j^t}{\theta_j^t(1)}$$

(9)

$$P_t = \frac{1}{N}\sum_{j=1}^{N} \frac{\theta_j^t}{SE(\theta^t)}$$

(10)

$$P_\alpha = \frac{P_a}{T}$$

(11)

In the equation, $G_t$ and $G_\alpha$ refer to group statistics, while $P_t$ and $P_\alpha$ refer to panel statistics.

### 3.2.5. Long-Run and Short-Run Estimations

This study employs the CS-ARDL estimator proposed by Chudik and Pesaran (2015) to examine the long-run and short-run coefficients. The CS-ARDL procedure offers several advantages over conventional econometric models, simultaneously addressing slope heterogeneity across countries and cross-country dependency, irrespective of whether the related variables are non-stationary I(0), stationary I(1), or of mixed-order integration. Furthermore, the CS-ARDL technique rectifies issues related to unobserved common factors, serial correlation, and correction bias, displaying resilience in handling endogeneity arising from reverse causal relationships among variables, while also tackling concerns related to small sample size and omitted variable biases. Additionally, a noteworthy aspect of the CS-ARDL mechanism is its ability to normalize the influences of unobserved common factors by incorporating the Pesaran (2006) corrected effects procedure within the framework of panel ARDL models. This involves considering the lagged dependent variable as a weakly exogenous variate under the error correction framework. Subsequently, the panel CS-ARDL estimation takes the following form:

$$\Delta Y_{it} = \varphi_i + \sum_{j=1}^{p} \overline{\varphi}_{it}\Delta Y_{i,t-1} + \sum_{j=0}^{q} \overline{\varphi}_{it}X_{i,t-1} + \sum_{j=1}^{r} \overline{\varphi}_{it}\overline{CS}_{,i,t-1} + \varepsilon_{i,t}$$

(12)

In the equation, *CS* presents the cross-sectional mean which is further expressed as ($\Delta Y_{it}$, $X_t$); $X$ presents regressors (i.e., $LREM_{it}$, $LRGDPP_{it}$, $LRES_{it}$, $LGLOB_{it}$, and $LFDI_{it}$), and $p$, $q$, $r$ represent the lags for each variable.

To examine the robustness and further validate the findings obtained through CS-ARDL, this study employed the Pesaran (2015) CCEMG and the Eberhardt and Bond (Eberhardt and Bond 2009) AMG estimation methods. Similar to CS-ARDL, these frameworks aim to address challenges related to common shocks, parameter endogeneity, unit root, and panel slope heterogeneity. The CCEMG estimator involves the linear arrangements of the cross-sectional mean of observable dependent and explanatory variables along with the common unobserved effects. On the other hand, the AMG estimator introduces a common dynamic effect in panel data models to accommodate cross-sectional dependency.

## 4. Results

This section of the study presents the findings of different econometric techniques discussed in the earlier section. The result of the Pesaran (2004) cross-sectional dependency test is presented in Table 4. The null hypothesis of cross-sectional independence is rejected at 1% level, implying cross-sectional dependence among the series. The slope heterogeneity test result based on Pesaran and Yamagata (2008) is presented in Table 5. The null hypothesis of slope homogeneity is rejected for both the tests (i.e., $\check{\Delta}$ and $\check{\Delta}_{adj}$) at 1% significant level, confirming the strong slope heterogeneity among the cross-sections. The result of the second-generation CIPS panel unit root test is presented in Table 6. Apparently, all the variables are non-stationary at this level, after considering the first difference, i.e., I(1) all panel variables become stationary.

Additionally, Table 7 displays the findings of the Westerlund cointegration test. The null hypothesis of no cointegration is rejected, specifically for three out of four test statistics, at a significance level of 1%. Consequently, this affirms the presence of a cointegration relationship among the variables.

After identifying the cointegration, we utilize the CS-ARDL estimator, and the outcomes of both the long-run and short-run results are presented in Table 8. The findings suggest a positive relationship between remittance flows and sustainable economic development. In both the long run and the short run, a 1% positive change in remittance flow is associated with an increase in sustainable economic development by 0.155% and 0.081%, respectively. Remittance emerges as a pivotal contributor in the form of financial capital, fostering capital accumulation within the recipient economy. This surplus capital, in turn, manifests itself in heightened productivity, increased income levels, and an overall augmentation of economic development over an extended timeframe.

The findings agree with the arguments put forth by Hien et al. (2020) supporting the notion that remittances, when directed towards investment and savings, can serve as a catalyst for sustainable economic growth. Hassan and Holmes (2013) noted that when remittance-to-GDP ratio was more than 1% in an economy, remittance was mainly diverted to the production sector. Our finding is also in line with Wang et al. (2021), Islam (2022), Li and Yang (2023) who noted that remittance helps to curb carbon emissions in the long run. This underscores the significance of remittances not only as a financial lifeline for individual households but also as a macroeconomic driver, stimulating sustainable economic development. Both in the short run and the long run, there may be a multiplier effect at play. The initial injection of remittances into the economy could lead to a series of additional spending and investment, amplifying the overall impact.

The coefficient of LRGDPP has a positive sign, revealing that a 1% rise in economic growth causes adjusted net savings to rise by 0.538% and 0.502% in the long and the short run, respectively. The finding is consistent with Din et al. (2021) and Hunjra et al. (2022b). However, it contrasts with Nchofoung and Asongu (2022). Our study is in line with the theory, that is, with an increase in per capita GDP, economies can divert more resources towards sustainable development projects. This may include investments in renewable energy, environmental conversation, and social development.

The impact of natural resource rent (LRES) on sustainable economic development is negative and statistically significant. A 1% positive change in natural resource rent decreases adjusted net savings by 0.084% and 0.087% in the long and the short run, respectively. This finding is not surprising as many existing studies on developing and emerging economies have found similar results (Atkinson and Hamilton 2003; Din et al. 2021; Sadik-Zada 2023; Van der Ploeg 2010). The results suggest that the rate of increase in natural resource exploitation surpasses the rate of increase in investment in human and physical capital. Therefore, economies should prioritize increasing the rate of investment in human and physical capital to achieve SED.

**Table 4.** Results of the Pesaran (2004) cross-sectional dependency test.

| Variables | Pesaran CD |
|:---:|:---:|
| LANS | 55.017 *** (0.000) |
| LREM | 114.708 *** (0.000) |
| LRGDPP | 93.417 *** (0.000) |
| LRES | 112.778 *** (0.000) |
| LGLOB | 92.121 *** (0.000) |
| LFDI | 92.252 *** (0.000) |

Note: *** denotes significance at 1% level. Probability values are in brackets.

**Table 5.** Results of the Pesaran and Yamagata (2008) slope heterogeneity test.

| Slope Homogeneity Test | |
|:---:|:---:|
| $\check{\Delta}$ | 26.148 *** (0.000) |
| $\check{\Delta}_{adj}$ | 30.587 *** (0.000) |

Note: *** denotes significance at 1% level. Probability values are in brackets.

**Table 6.** Results of the CIPS unit root test.

| | Level | | First Difference | | |
|:---:|:---:|:---:|:---:|:---:|:---:|
| Variables | constant | constant with trend | constant | constant with trend | order |
| LANS | −2.067 * | −2.427 | −4.910 *** | −5.010 *** | I(1) |
| LREM | −1.999 | −2.481 * | −4.510 *** | −5.572 *** | I(1) |
| LRGDPP | −1.499 | −1.560 | −3.269 *** | −3.601 *** | I(1) |
| LRES | −1.897 | −2.221 | −4.632 *** | −5.650 *** | I(1) |
| LGLOB | −1.944 | −2.114 | −4.898 *** | −4.997 *** | I(1) |
| LFDI | −2.015 | −2.219 | −5.330 *** | −5.415 *** | I(1) |

Note: ***, *denote significance at 1%, and 10% levels.

**Table 7.** Results of the Westerlund cointegration test.

| Test Statistics | Value | Z-Value | *p*-Value |
|:---:|:---:|:---:|:---:|
| $G_t$ | −3.157 *** | −7.070 | 0.000 |
| $G_a$ | −11.415 | −0.323 | 0.323 |
| $P_t$ | −28.678 *** | −14.032 | 0.000 |
| $P_a$ | −18.781 *** | −12.450 | 0.000 |

Note: *** denotes significance at 1% level.

**Table 8.** Results of CS-ARDL in the long run and the short run.

| Variables | Long-Run Results | | |
|:---:|:---:|:---:|:---:|
| | Coefficients | Standard error | Probability |
| LREM | 0.155 ** | 0.079 | 0.049 |
| LRGDPP | 0.538 * | 0.311 | 0.084 |
| LRES | −0.084 * | 0.047 | 0.073 |
| LGLOB | 1.422 *** | 0.450 | 0.000 |
| LFDI | 0.044 | 0.354 | 0.901 |
| | Short-Run Results | | |
| LREM | 0.081 * | 0.048 | 0.094 |
| LRGDPP | 0.502 ** | 0.222 | 0.024 |
| LRES | −0.087 * | 0.052 | 0.096 |
| LGLOB | 1.933 *** | 0.805 | 0.000 |
| LFDI | −0.082 | 0.213 | 0.700 |
| ECT $_{(-1)}$ | −0.929 *** | 0.051 | 0.000 |

Note: ***, **, * denote significance at 1%, 5%, and 10% levels.

Globalization (LGLOB) has indicated a positive and significant impact on sustainable economic development both in the long and the short run, whereby a 1% rise in globaliza-

tion will increase adjusted net savings by 1.422% and 1.933%, respectively. These findings are consistent with those of Arif et al. (2022) and support the underlying theory that, when managed effectively, globalization has the potential to contribute to sustainable economic development by fostering interconnectedness among economies, promoting the efficient allocation of resources, and facilitating the transfer of knowledge and technology.

The coefficient of foreign direct investment (LFDI) is positive in the long run and negative in the short run; however, the result does not exhibit any statistical significance. This outcome aligns with our initial hypothesis, emphasizing the nuanced and ambiguous nature of FDI's impact on sustainable development, underscoring the complexity of economic interactions over varying time horizons. Wang et al. (2023) also found an insignificant impact of FDI on sustainable development.

We use AMG and CCEMG estimators for robustness analysis, and the result is presented in Table 9. The direction of coefficient and the significance level are almost the same. Subsequently, we can confirm that remittances, real GDP per capita, globalization, and natural resource rents influence sustainable development in developing and emerging economies.

**Table 9.** Long-run robustness results of AMG and CCEMG.

| Variables | AMG | | |
|---|---|---|---|
| | Coefficients | Standard error | Probability |
| LREM | 0.064 *** | 0.018 | 0.000 |
| LRGDPP | 0.579 *** | 0.131 | 0.000 |
| LRES | −0.042 *** | 0.015 | 0.000 |
| LGLOB | 0.164 * | 0.096 | 0.090 |
| LFDI | −0.032 | 0.105 | −0.31 |
| | CCEMG | | |
| LREM | 0.080 ** | 0.024 | 0.001 |
| LRGDPP | 0.726 *** | 0.194 | 0.000 |
| LRES | −0.035 ** | 0.015 | 0.022 |
| LGLOB | 0.138 * | 0.081 | 0.071 |
| LFDI | 0.019 | 0.030 | 0.516 |

Note: ***, **, * denote significance at 1%, 5% and 10% levels.

## 5. Conclusions

The core goal of this study is to analyze the impact of remittance inflow on sustainable economic development in the case of 52 high remittance-receiving developing and emerging economies from 1996 to 2021. The dependent variable is adjusted net savings (ANS), which is used as a proxy for sustainable economic development. The independent variables include remittance, real GDP per capita, total natural resource rent, globalization, and foreign direct investment. First, the study checked for cross-sectional dependency and employed the second-generation panel unit root test. Second, the study adopted the second-generation cointegration test. Third, the study used CS-ARDL estimators for long-run and short-run estimates. Fourth, the study employed AMG and CCEMG estimators for assessing the robustness of long-run estimates of the variables. The study noted several key findings that have important policy implication.

Firstly, we noted that remittances contribute to SED, which is measured by ANS. The underlying mechanism is that remittances are largely invested in education, physical capital, agricultural productivity improvement, and green projects. It is also likely that remittances are used to improve energy efficiency and thus reduce $CO_2$ emissions. Thus, remittance inflow is aiding developing countries' future generations to have more productive resources to meet their economic needs. The policy implication is that policymakers should implement policies to facilitate and incentivize remittance flows, such as reducing transaction costs and creating favorable exchange rate regimes. In addition, policymakers should develop programs that educate diaspora communities on investment opportunities

in their home countries to encourage a more sustainable use of remittances. However, the magnitude of the impact of remittances on sustainable development should be interpreted with caution as this study did not consider the loss of human capital through migration.

Secondly, the study noted the positive impact of globalization on SED. This suggests that policymakers should continue to pursue policies that promote international trade and economic integration to achieve both short- and long-term SED. Thirdly, the study noted the positive contribution of per capita income on SED. Therefore, the less developed countries should actively pursue policies that promote short- and long-term per capita income. This finding further reinforces the role of investment in education, skill development, and healthcare and as physical capital as it is not only related to GDP but also SED. Fourthly, we did not find any evidence of the effect of FDI on SED, and this suggests that, under the current context, there is lack of FDI directed towards building long-term economic capacity of countries.

Finally, and very importantly, the study noted that natural resource rent negatively contributes to SED. The finding suggests that resource rent is not sufficiently reinvested into human and physical capital, and thus, reinvestment is not sufficient to compensate for the depletion of the natural resources. Thus, with other things held constant, natural resources extraction is likely to reduce the amount of productive assets available to future generations to meet their economic needs. The policy implication of these findings is that policymakers should implement strict regulations and sustainable practices in the extraction and utilization of natural resources to minimize their negative impact on economic development. Moreover, the increasing share of resource rent should be directed towards human and capital accumulation.

In summary, similar to other studies, this study has some limitations that provide us with future study guidelines. We selected developing and emerging economies; in a future study, we will use these variables to explore regional heterogeneity. In this study, we did not use any moderating factor, but in a future study, we may look into interactive effects such as institutional quality and remittance effects on sustainable economic development. We employed CS-ARDL, AMG, and CCEMG, and in a future study, we will look into non-linear analysis.

**Author Contributions:** Conceptualization, S.A.C. and B.S.; methodology, S.A.C.; software, S.A.C. and B.S.; validation, S.A.C. and B.S.; formal analysis, S.A.C. and B.S.; investigation, S.A.C. and B.S.; resources, S.A.C.; data curation, S.A.C. writing—original draft preparation, S.A.C. and B.S.; writing—review and editing, S.A.C. and B.S.; visualization, S.A.C. and B.S.; supervision, B.S.; project administration, S.C; All authors have read and agreed to the published version of the manuscript.

**Funding:** There was no external funding received for this project.

**Data Availability Statement:** All data is publicly available, and at the time of writing and publication, the data were accessible from the respective websites mentioned in the Data section of the paper.

**Acknowledgments:** Both the authors sincerely thank the editors and the anonymous reviewers for their useful suggestions and recommendations; all remaining errors are ours. Shasnil Avinesh Chand sincerely acknowledges the financial support of the University of the South Pacific research office.

**Conflicts of Interest:** The authors declare no conflict of interest.

## Appendix A

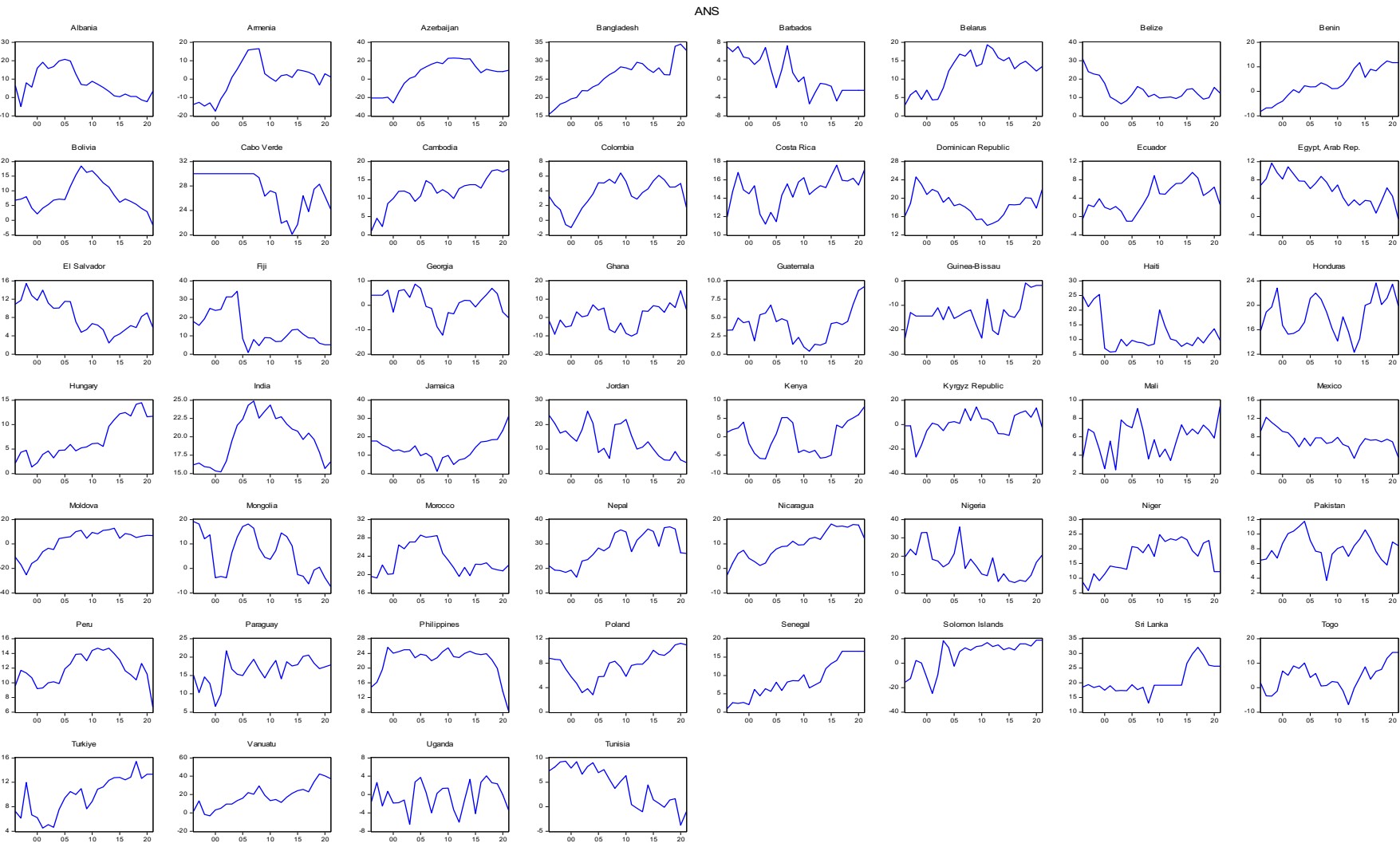

**Figure A1.** Adjusted net savings (% GNI).

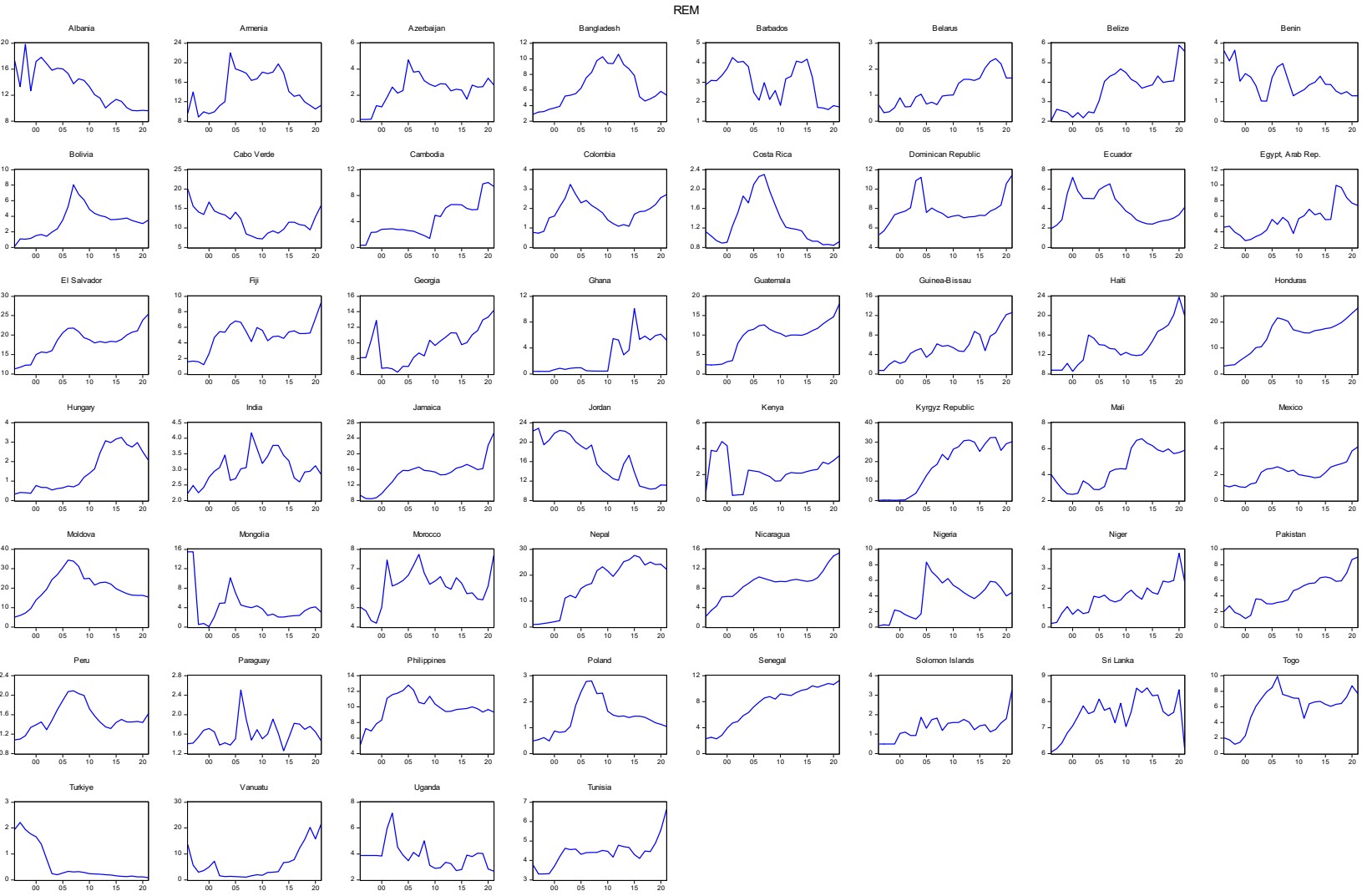

**Figure A2.** Total remittance received (% GDP).

## Note

1   The authors conducted the calculation based on the World Bank data from 2023. This involved adding up the total remittances received by 52 developing and emerging economies in USD for the year 2021. The next step was to divide this sum by the total global remittances received in USD for the same year. The result was then multiplied by 100. The data on macroeconomic variables are attained from the World Development Indictors (World Bank 2024). The globalization index is obtained from the KOF globalization index data (https://datafinder.qog.gu.se/dataset/dr) (assessed on 27 February 2024) (c.f. Dreher 2006; Dreher et al. 2008). The definitions of the variable used in this study are provided in Table 1. The dependent variable is adjusted net savings (ANS) as the percentage of gross national income (GNI) (Din et al. 2021; Hunjra et al. 2022a; Hussain et al. 2023). The descriptive statistics and the correlation matrix for the data used are provided in Tables 2 and 3

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
