# Peer review of "Role of Remittance on Sustainable Economic Development in Developing and Emerging Economies: New Insights from Panel Cross-Sectional Augmented Autoregressive Distributed Lag Approach"

_jrfm, doi:10.3390/jrfm17040153_

Round 1

Reviewer 1 Report

Comments and Suggestions for Authors

I have some suggestions that is useful to improve the quality of this paper.

1. In section 1. Introduction, page 4:” Accordingly, this study aims to … ANS is a recently developed proxy for assessing economic sustainability.” Can the authors give examples of literature where ANS as an indicator for assessing economic sustainability?

2. In section 2. Literature Review, the structure of this section is not clear enough and lacks a summarizing statement. Should the literature on sustainable economic development not be limited to carbon emissions? Could the authors reorganize the section?

3. In section 3.1 Data:

(1) The first paragraph lists the names of 53 countries, with the same country name ”Tunisia” appears twice. After checking with the country names in the Figure A1 in Appendix A, some of the country names are incorrect: Jordon should be Jordan; Tukey should be Turkiye.

(2) The authors state that ”The descriptive statistics and correlation matrix for the data used are provided in Table 2, and in Table 3.” Actually, both the descriptive statistics and correlation matrix are on Table 2, and the correlation matrix table lacks horizontal vectors. In addition, can the authors provide a simple analysis of the coefficient matrix table?

4. In section 3.2 Method:

(1) In section 3.2.1 Cross-Section dependence test, Can the authors explain the meaning and purpose of this test in the text? Also, can the authors explain the meaning of the variables in the equation(3)?

(2) In section 3.2.5 Long-run and short-run estimation, should the authors change Xu,i,t-1 to Xi,t-1 in the model(12)? Can the author explain the meaning of the variables j, q, y, w in the model(12)?

5. In section 4. Results:

(1) “The coefficient of foreign direct investment (LFDI) …, This outcome aligns with our initial hypothesis…”. This conclusion is inconsistent with the previous 3.2 Method (iv) hypothesize, which assumes that foreign direct investment has positive effect on sustainable economic development.

(2) Can the authors elaborate on Model 1 and Model 2 in Table 4, which are not represented in the previous text?

Comments on the Quality of English Language

Minor editing of English language required

Reviewer 2 Report

Comments and Suggestions for Authors

I am pleased to review the manuscript titled “Role of Remittance on Sustainable Economic Development in Developing and Emerging Economies: New Insights from the Panel CS-ARDL Approach.” This research is intriguing and contributes significantly to the field, marking the first study to explore the impact of remittances on adjusted net savings, which serves as an indicator of sustainable economic development. However, I have certain comments for consideration:

1. Theoretical Foundation Clarification Needed: While the authors note that their methodology aligns with the standard literature on remittances and growth, it is crucial to specify the underlying theory and elucidate the mechanism. This clarification should include proper citations to support the anticipation that remittances might have a positive impact, despite the literature review indicating a lack of consensus on remittances' effects. Additionally, once a solid theoretical foundation is established, a hypothesis regarding the remittances' effect on sustainable economic development should be formulated.

2. Literature Review Revision Required: The current literature review merely summarizes previous research findings, lacking in-depth justification for the empirical results. This section should be restructured to explain the underlying reasons behind the varied impacts. For example, how to explain positive, negative, and insignificant effects of remittances on economic growth, rather than simply listing the findings. These also include other dependent variables such as carbon emissions, etc.

3. Rationale Behind Select Control Variables Needs Strengthening:

3.1. Natural Resource Rents: Can the Dutch Disease affect sustainable economic development? How might an increase in the exports of natural resources impact sustainable economic development negatively? These questions require a clear explanation to understand the implications of the Dutch Disease on an economy's long-term growth and environmental sustainability. The manuscript should delve into the mechanisms by which the Dutch Disease could potentially hinder sustainable economic development, considering both the economic and environmental dimensions.

3.2. Foreign Direct Investment (FDI): While the authors posit that FDI positively affects sustainable economic development, fundamental economic theories—such as the pollution halo hypothesis and the pollution haven hypothesis—suggest that FDI's impact on the environment (and consequently on sustainable development) can vary. This section requires a thorough revision to reflect these complexities.

Comments on the Quality of English Language

Fine

Round 2

Reviewer 1 Report

Comments and Suggestions for Authors

The detailed response to my last report is satisfactory, I thus recommend acceptance of the revised manuscript.

Comments on the Quality of English Language

Minor editing of English language required.